# Magnetic Field Alignment, a Perspective in the Engineering of Collagen-Silica Composite Biomaterials

**DOI:** 10.3390/biom11050749

**Published:** 2021-05-18

**Authors:** Nicolas Debons, Kenta Matsumoto, Noriyuki Hirota, Thibaud Coradin, Toshiyuki Ikoma, Carole Aimé

**Affiliations:** 1Laboratoire de Chimie de la Matière Condensée de Paris (LCMCP), Sorbonne Université, CNRS, 75005 Paris, France; debons.nicolas@hotmail.fr (N.D.); thibaud.coradin@sorbonne-universite.fr (T.C.); 2Tokyo Institute of Technology, School of Materials and Chemical Technology, Department of Materials Science and Engineering, Ookayama 2-12-1, Meguro-ku, Tokyo 152-8550, Japan; matsumoto.k.be@m.titech.ac.jp (K.M.); tikoma@ceram.titech.ac.jp (T.I.); 3National Institute for Materials Science, Fine Particles Engineering Group, 3-13 Sakura, Tuskuba 305-0003, Japan; hirota.noriyuki@nims.go.jp; 4Ecole Normale Supérieure, CNRS-ENS-SU UMR 8640, 24 rue Lhomond, 75005 Paris, France

**Keywords:** bionanocomposites, collagen, high magnetic field, silica particles

## Abstract

Major progress in the field of regenerative medicine is expected from the design of artificial scaffolds that mimic both the structural and functional properties of the ECM. The bionanocomposites approach is particularly well fitted to meet this challenge as it can combine ECM-based matrices and colloidal carriers of biological cues that regulate cell behavior. Here we have prepared bionanocomposites under high magnetic field from tilapia fish scale collagen and multifunctional silica nanoparticles (SiNPs). We show that scaffolding cues (collagen), multiple display of signaling peptides (SiNPs) and control over the global structuration (magnetic field) can be combined into a unique bionanocomposite for the engineering of biomaterials with improved cell performances.

## 1. Introduction

Bionanocomposites include different types of materials with common features that are the combination of elements of different chemical nature, one being of biological origin. Most importantly, their development is part of a strategy aimed at synergistically improving their physical and chemical stability as well as their functionality, including their interface with cells and living systems [1,2]. Our research focuses on the design of biomimetic fibrillary extracellular matrices (ECMs) for guiding cells in regenerative contexts. While this may include biological or synthetic polymers [3], we focus here on type I collagen, which is of particular relevance for designing biomimetic ECMs as it is a major constituent of connective tissues [4,5]. Many different processes have been used to shape collagen while preserving its native conformation [6], including extrusion [7], aerosols [8], electrospinning [9] and freeze-drying technologies [10]. High magnetic field has also been used to control collagen alignment, as collagen fibrils orient perpendicularly to the field direction [11,12,13,14]. This behavior can be explained by the torque induced by magnetic field in the planar peptide bonds of collagen due to their diamagnetic anisotropy, Δχ. With a constant magnetic field B, the total energy needed for collagen fibril orientation can be noted E1 = VΔχB^2^/(2µ_0_), with V the volume of self-assembled collagen fibrils and µ_0_ the permeability of vacuum, while thermal energy, i.e., Brownian motion, can summarily be noted E2 = k_B_T, with k_B_ the Boltzmann’s constant, and T the temperature. When E1 becomes superior to E2 (VΔχH^2^/(2µ_0_) > k_B_T), or to be more accurate, to the total internal energy of collagen solution, then controlled alignment of collagen fibrils can be achieved. Moreover, as collagen fibrils spontaneously self-assemble into thicker bundles, V and therefore E1 keeps increasing and their alignment intensifies [15].

Previous works showed the possibility to tune the orienting effect of magnetic field throughout the anisotropic hydrogel over a given fibrillogenic period, playing with temperature, collagen concentration and magnetic strength [15,16]. Twenty years after their discovery, Torbet et al. successfully oriented magnetically collagen fibrils to develop an assembled plywood structure as an artificial scaffold for corneal transplantation [17]. Since that time, several works in the literature reported the possibility to use magnetic orientation for the engineering of tissues naturally showing fibril alignment, including bone, tendon and nervous tissues [18,19,20,21]. Magnetically-oriented ECM-like materials also find a high relevance in biomedical research given that aligned collagen scaffolding is one feature of the tumor ECM [22,23].

Composite approaches have been developed by mixing collagen with paramagnetic iron oxide beads, where the particles were found to orient under low magnetic field, resulting in collagen fibril alignment [24]. Here we design bionanocomposites from non-magnetic silica nanoparticles (SiNPs) where the orientation of collagen fibers relies on high magnetic field. SiNPs are used owing to their versatile chemistry that allows conjugating a broad diversity of biologically-relevant functional groups. We have previously shown that engineering the SiNP surface affects the scaffold structure and ultimately impacts cell response [25,26]. Beyond scaffolding, biochemical signaling can be provided by the conjugation of selected peptide epitopes. Integrin receptors are a common target for ensuring cell adhesion to biomaterials. This includes the RGDS peptide sequence within fibronectin, which synergizes with the PHSRN sequence in a distance-dependent manner, substantially enhancing cell adhesion mediated by the α5 β1 integrin receptor for fibronectin [27,28]. We have recently reported that surface modification of SiNPs can be successfully used to simultaneously display RGDS and PHSRN enabling enhanced cell adhesion and spreading [29]. Significant advances for regenerative medicine can be expected by coupling signaling abilities with fine control over structure. In this context, we report here the design of bionanocomposites from tilapia fish scale collagen and functionalized SiNPs prepared under high magnetic field. We show that dynamic control of collagen fibrillogenesis under magnetic field is a robust process that can accommodate bioconjugated nano-objects and achieve their controlled dispersion while preserving a global anisotropic structure of the composite. Most importantly, being fully compatible with cell studies, this process is shown to preserve the ability of the composite—via the use of multifunctional SiNPs—to interact with cells and enhance their adhesion and spreading.

## 2. Materials and Methods

### 2.1. Materials

For the synthesis and functionalization of SiNPs, tetraethyl orthosilicate (TEOS 98%), (3-Mercaptopropyl)trimethoxysilane (MPTMS 95%), and sulfuric acid (H_2_SO_4_) were purchased from Sigma-Aldrich (France). Absolute ethanol (GPR RectaPur, VWR, France) was purchased from VWR, ammonium hydroxide solution (25%) from Carlo Erba (France) and hydrogen peroxide (H_2_O_2_ 35%) from Acros Organics (France). For the peptide coupling, hexafluorophosphate benzotriazole tetramethyl uranium (HBTU, 98%) was purchased from Iris Biotech GmbH (Marktredwitz, Germany). N,N-diisopropylethylamine (DIEA, 99.5%) and trifluoroacetic acid (TFA) were purchased from Sigma-Aldrich and dichloromethane (DCM) from Carlo Erba.

Freeze-dried type I collagen from tilapia fish-scale was purchased from Taki Chemical Co., Ltd. (Japan). All culture reagents were purchased from Gibco (ThermoFisher Scientific, France)). AlexaFluor-488-conjugated phalloidin and DAPI were purchased from Life Technologies.

### 2.2. Synthesis of Functionalized SiNPs

#### 2.2.1. Synthesis of nf-SiNPs

According to the Stöber procedure [30], 21 mL TEOS was added dropwise to a solution containing 32 mL ultrapure water, 600 mL absolute ethanol and 45 mL ammonium hydroxide solution. The solution was left under stirring overnight at RT. SiNPs were washed with ethanol through an ultrasonic redispersion-centrifugation process (10,000 rpm for 5 min) and dried overnight at 30 °C under vacuum. Size of the resulting *nf*-SiNPs was 214 ± 10 nm, as determined by transmission electron microscopy (see below).

#### 2.2.2. Synthesis of SiNP-SO_3_^−^

*nf*-SiNPs were first functionalized with thiol groups by silylation with MPTMS. Typically, 4.12 g of silica particles were redispersed in a mixture of 410 mL absolute ethanol and 9.1 mL NH_4_OH before addition of 3.2 mL MPTMS. The mixture was stirred for 40 min at RT. Then, the reaction mixture was heated to 80 °C until evaporation of the two thirds of the volume. The mixture was left to cool down to RT and subsequently washed three times with absolute ethanol through an ultrasonic redispersion-centrifugation process (12,000 rpm for 15 min) and dried under vacuum. Finally, oxidation of thiol groups leads to sulfonic acid functionalized particles [31]. In a typical reaction, 0.6 g of thiol-modified particles were suspended in 30 mL H_2_O_2_ under stirring at RT for 48 h. The powder was washed by centrifugation before addition of 25 mL H_2_SO_4_ and stirred for 2 h at RT. Resulting SiNP-SO_3_^−^ particles were washed with ethanol and water through an ultrasonic redispersion-centrifugation process (10,000 rpm for 5 min) before drying under vacuum at 30 °C.

#### 2.2.3. Synthesis of Amine-Modified Particles SiNP-NH_2_ (for Peptide Conjugation)

*nf*-SiNPs were modified with amine groups thanks to 3-aminopropyltriethoxysilane (APTES). Typically, 0.77 g of *nf*-SiNPs were dispersed in a mixture of 76.6 mL ethanol and 1.7 mL ammonium hydroxide solution before addition of 0.75 mL APTES. The mixture was stirred for 2 h at RT. Then, the reaction mixture was heated to 80 °C until evaporation of the two thirds of the volume. The mixture was left to cool down to RT and subsequently washed three times with absolute ethanol through an ultrasonic redispersion-centrifugation process (10,000 rpm for 5 min) before drying the obtained SiNP-NH_2_ particles under vacuum at 30 °C.

#### 2.2.4. Peptide Coupling between Amine Groups on SiNPs and Peptides

The RGDS and PHSRN peptides were synthesized and purified at the Protein Engineering and Q-PCR Platform of the Institut de Biologie Paris-Seine (Sorbonne Université, Paris, France) by C. Piesse. A rink amide resin was used, with the classical Fmoc-strategy on a 0.2 mM scale. The peptide PHSRN was cleaved from the resin with deprotection of all protective groups, which was not the case of RGDS because of a pending carboxylic group due to the presence of aspartic acid. Peptide conjugation of RGDS, PHSRN or a stoichiometric 1:1 mixture of the two was performed between final carboxylic acid on RGDS or PHSRN and amine groups at the surface of SiNP-NH_2_. Typically, 33.4 mg of SiNP-NH_2_ (5.1.10-4 mmol) were dispersed in a mixture of 140 μL DMSO and 34 μL DMF, with 4 equivalents of peptide, HOBT.H_2_O and HBTU and 8 equivalents of DIEA. The mixture was stirred for 40 min and subsequently washed twice with DMSO and twice with DCM through an ultrasonic redispersion-centrifugation process (10,000 rpm for 5 min) before drying under vacuum. Ultrasonication was performed before centrifugation for *ca* 1 min at RT until obtention of a homogeneous particle suspension using an Elma S30 Elmasonic instrument working at an ultrasonic frequency of 37 kHz and an ultrasonic power effective of 80 W. To deprotect the amine from the BOC group, an acidic treatment with TFA was performed under sonication for 20 min. SiNP-RGDS and SiNP-PHSRN were washed three times with absolute ethanol through an ultrasonic redispersion-centrifugation process (10,000 rpm for 5 min) before drying under vacuum.

### 2.3. Particle Characterization

#### 2.3.1. Zeta-Potential Measurements

Zeta-potential measurements were performed on a Malvern Zetasizer Nano spectrometer (Malvern Panalytical, Malvern, UK), at a concentration of particle of 0.5 g.L^−1^ in DTS1060C cells in 100 mM KCl buffer at different pHs adjusted with NaOH and HCl at RT.

#### 2.3.2. Transmission Electron Microscopy

A drop of sample in aqueous solution was deposited on carbon-coated copper grids (300 mesh). After 3 min, the excess liquid was blotted with filter paper. TEM was performed at RT using a Tecnai spirit G2 electron microscope operating at 120 kV and the images were recorded on a Gatan Orius CCD camera (Gatan, Pleasanton, CA, USA).

### 2.4. Preparation of the Bionanocomposites

Freeze-dried type I collagen from tilapia fish-scale was prepared at 0.62 wt% in HCl 0.64 mM [32].

For the preparation of collagen-SiNP bionanocomposites, SiNPs were first weighted, diluted in ethanol and put under sonication (Elma S30 Elmasonic instrument working at an ultrasonic frequency of 37 kHz and an ultrasonic power effective of 80 W) until obtention of a homogeneous suspension. The SiNP suspension was added to the acidic solution of collagen under stirring so as to reach a final SiNP concentration of 5 mg.mL^−1^. After depositing collagen or collagen-SiNPs suspension and PBS in a plastic pot, a planetary centrifugal bubble free mixer (Thinky Inc., Laguna Hills, CA, USA.) was used during 30 s at 2000 rpm to mix collagen or collagen-SiNPs and PBS solutions, and 30 s at 2200 rpm to degas the viscous mixture (Figure 1). Then, a Microman micropipette was used to introduce 800 µL of the resulting solution in the silicone rubber mold between two slide glasses. Six silicone molds were introduced in a suitable plastic beaker. Three of these beakers were placed on the three slots for stirring on the bench (Figure 2). Care was taken towards the best alignment of the beakers along the direction of magnetic field, i.e., along the cylindrical hole inside the magnet. The 60 rpm rotation was tested before starting experiment thanks to a VEXTA OPX-1 motor (Oriental Motors), and the bench was introduced into the JASTEC superconductor 13 T magnet until the middle beaker is located at 282 mm from the entrance, where the magnetic field is actually equal to 13 T (both other beakers face a 12 T magnetic field). Before preparing the composite solutions, the thermostat made of 28 °C water-filled tubes was introduced inside the magnet and water heating was turned on.

### 2.5. Characterization of the Structural and Biological Properties of the Bionanocomposites

#### 2.5.1. Scanning Electron Microscopy (SEM)

Composite membranes made of magnetically aligned collagen fibrils were deposited on carbon-tape coated aluminum pads. Samples were coated with a 10 nm gold layer before observations. SEM imaging was performed using a Variable Pressure Hitachi S-3400 N working at an accelerating voltage of 10 kV, using an in lens secondary electron (SE) detector, with a working distance ca. 4.2 mm.

#### 2.5.2. Polarized Light Microscopy (PLM)

PLM was performed using a transmission Nikon Eclipse E600 Pol, equipped with crossed polarizers, a waveplate and a Nikon DXM 1200CCD camera.

#### 2.5.3. Cell Culture

Composites underwent a gamma-ray treatment of 5 h for sterilization. A crosslinking carbodiimide treatment is then applied to the membranes, based on a 24 h soaking in 1% *w*/*v* 1-ethyl-3-(3-dimethylaminopropyl)carbodiimide hydrochloride (EDC.HCl) in PBS solution, before 5 washings in PBS.

The 3T3 cells were maintained in growth medium containing Dulbecco’s Modified Eagle’s Medium (DMEM), supplemented with 10% fetal bovine serum (FBS), 1% glutamax, 1% fungizone and 1% penicillin-streptomycin (P/S). The cells were grown in 75 mm^2^ flasks (BD Falcon), kept at 37 °C in a 5% CO_2_ atmosphere and passaged every three days. Before confluence, cells were removed from culture flasks by treatment with 0.1% trypsin and 0.02% EDTA. Cells were rinsed and suspended in complete culture medium before use. For cell morphology experiments on membranes, fibroblasts were seeded at a low density (5000 cells per well) in order to minimize cell–cell contacts. Cells were incubated at 37 °C and 5% CO_2_ for 24 h on the membranes into a 48-well plate. Cell culture experiments were performed in triplicates (*n* = 3).

#### 2.5.4. Fluorescence Microscopy

For fluorescence imaging, cells were fixed with 4% paraformaldehyde with 1 mM CaCl_2_ in PBS at RT for 10 min. For immunostaining, fixed samples were first permeabilized with 0.1% Triton X-100 in PBS (20 min, RT). Actin filaments were fluorescently labeled with AlexaFluor-488-conjugated phalloidin (165 nM, 1 h at RT in a dark chamber) for visualization. Cell nuclei were stained with DAPI (300 nM, 10 min at RT in a dark chamber). Samples were analyzed using an Axio Imager D.1, Zeiss fluorescence microscope (Oberkochen, Germany).

Cell metabolic activity was monitored at 24 h using Alamar Blue assay. Fibroblasts were incubated with 200 µL of a resazurin solution at 0.01% (*w*/*v*) for 3 h. The supernatant in each well was then collected and the absorbance measured at λ = 570 nm and 600 nm. The percentage of resazurin reduction was calculated following the formula provided by the supplier. Cell metabolic activity was compared to control samples, i.e., cells cultivated without membranes. The arbitrary value of 100% was given to control samples.

Cell number was quantified from phalloidin and DAPI stained fluorescent images acquired by a 4× objective along a randomly selected 4 mm^2^ composite surface.

Cell morphology was quantified from phalloidin stained fluorescent images acquired by a 10× objective from randomly selected regions on the coverslip (at least 100 cells analyzed on each membrane). Acquired images were analyzed using ImageJ software.

#### 2.5.5. Statistical Analysis

Statistical analysis was performed on Graphpad Prism v.6 software (GraphPad Software, San Diego, CA, USA) using a Mann–Whitney non parametric test; each condition was tested from three independent experiments. Values in graphs are the mean and standard error of mean (SEM).

Collagen fiber orientation was determined using ImageJ software by measuring the angle of a length of 1-µm fiber with respect to the trigonometric circle. The resulting standard deviations for each condition (at least 20 measurements for at least 3 pictures for each condition) were determined and compared using Graphpad software. For low means, we concluded that the fibers had a better global orientation towards one given direction.

## 3. Results

### 3.1. Biocomposite Engineering

#### 3.1.1. Obtaining Self-Supported Collagen Films

Collagen-based gels were prepared in PBS 10× with a concentration (6.2 mg.mL^−1^ in HCl 0.64 mM) selected to ensure gel formation in the time course of high magnetic field exposure (6 h). Collagen and PBS were deposited at the bottom of a plastic pot in a 9:1 *v*:*v* ratio (Figure 1a), and mixing with planetary centrifugation was implemented to ensure the formation of a homogeneous gel and avoid air bubbles (Figure 1b). The resulting viscous mixture was transferred into a silicon mold (Figure 1c). After dehydration, a self-supported film was obtained, with a thickness of ca. 40 µm showing a homogeneous entangled collagen fibril network (Figure 1d).

For the exposition of samples to high magnetic field, the silicone mold was placed between glass slides (Figure 2a). Collagen fibrils were expected to align perpendicularly to the high magnetic field direction (y axis) i.e., in the xz plan (Figure 2b). Thus, in order to favor fibril alignment along only one axis, rotation of the mold around the z axis was additionally performed (Figure 2c). Gels were thus placed on a sample holder with three positions with independent rotations and introduced into the magnet, which was covered with water filled-tubes playing the role of a thermostat (Figure 2d,e). A magnetic field of 12 T was applied after investigations of different field strengths and exposure times, all ending up with materials exhibiting the 67-nm D-periodic structure specific to native collagen, and with adjustable properties [33]. We here selected a magnetic field of 12 T. As we have reported previously, in these systems collagen bundles with better-aligned fibril structures are obtained compared to what can be achieved with lower magnetic intensity (typically 6T) [13].

Finally, collagen hydrogels were dehydrated and crosslinked by an ethyl-dimethylcarbodiimide (EDC) 1% *w*/*v* solution in PBS. Fibril crosslinking prior or after magnetic field exposure had no influence on the resulting viscoelastic properties of the gels, indicating that crosslinking takes place within the collagen fibril and not between two fibrils, and enhancing fibril mechanical properties [13,34]. Polarized light microscopy (PLM) showed that EDC treatment enhances the birefringence of the collagen films indicating that the cross-linking favored fiber alignment (Appendix A).

#### 3.1.2. Surface-Engineered SiNPs

SiNPs, 214 nm in size (Figure 3a), were synthesized and added to the collagen solution prior gel formation. The surface chemistry of SiNPs has then been varied to impact (1) the scaffold structure and/or (2) the signal transduction abilities of the biocomposite. For the former, sulfonate-modified SiNPs (SiNP-SO_3_^−^) were selected for their strong electrostatic interactions with positively-charged soluble triple helices of collagen [25]. For the signal transduction, we focused on the clustering of bioactive signals onto SiNP surface by conjugating two integrin-binding peptides, RGDS and PHSRN, which are known to improve cell adhesion in a synergistic manner (Figure 3b) [27,28]. Synergistic interactions are known to be effective only for a peptide inter-distance of ca. 5 nm, which can happen when both peptides are statistically distributed at SiNP surface [29]. To fulfill this requirement, SiNPs were synthesized bearing one of these two peptide sequences (SiNP-RGDS, SiNP-PHSRN), or bifunctionalized with both peptides (SiNP-RGDS-PHSRN) (Figure 3c).

The functionalization of SiNPs was checked by zeta-potential measurements (Figure 3d). Non-functionalized particles (*nf*-SiNPs) showed neutral zeta potentials at low pH and negative values from pH 3–4 that reflect the presence of silanol groups (pKa ca. 3.5). The presence of sulfonate groups was confirmed as the zeta potential of SiNP-SO_3_^−^ was ca. −30 mV at all pHs, in agreement with the pKa of alkyl sulfonic acids (*ca.* 1). Zeta potentials of peptide-conjugated SiNPs were highly positive (between +20 and +30 mV) in acidic pHs, in agreement with the fact that RGDS bears one positive charge (arginine side group, pKa ca. 12) and PHSRN two positive charges (arginine and histidine side groups, pKa ca. 5) in such conditions. At pH 7 and above, zeta potentials start to decrease for all peptide-conjugated SiNPs, with a more negative slope for SiNPs presenting RGDS at their surface. This is consistent with the presence of a negative charge due to aspartic acid side chain of RGDS (pKa ca. 4). The positive values measured in basic conditions suggest that some unreacted amine groups are still present at the surface.

### 3.2. Birefringence of the Composites and Collagen Alignment

Polarized light microscopy (PLM) was used for its sensitivity to the birefringence of materials reflecting aligned structures. Series of images were acquired with multiple sample orientations relative to the crossed polarizers separated by 45°. In this configuration, a variation of contrasts, i.e., transmitted light, for different polarizer orientations would sign for a structural anisotropy of the materials: the composite interacts with light in an angle-dependent manner. In contrary, isotropic materials interact with light independently of the orientations of the polarizer.

#### 3.2.1. Collagen and Composite Films in Absence of Peptides

The pure collagen films showed a homogeneous birefringence characterized by a strong contrast when comparing images acquired with different polarizer orientations (Figure 4a1,a2). This reflects the alignment of collagen fibrils after high magnetic field exposure. The two biocomposites which do not incorporate peptides (collagen + *nf*-SiNP) and (collagen + SiNP-SO_3_^−^), exhibited numerous black spots, that we attribute to the presence of silica particles aggregates. In presence of *nf*-SiNPs, an important area of the sample seems not to change contrast with polarizer orientation (delimited by white dotted lines, Figure 4b1,b2), signing for a local disorganization of the collagen network.

In presence of SiNP-SO_3_^−^, no difference of contrast between the two polarizer orientations was evidenced (Figure 4c1,c2). This indicates a more general loss of collagen network organization upon introduction of these particles.

#### 3.2.2. Collagen: Peptide Composites

For all composites embedding peptide-conjugated SiNPs, no black dots similar to the above-described ones could be evidenced, suggesting the absence of extended SiNP aggregation. For (collagen + SiNP-RGDS) films, the contrast between two orientations of the crossed polarizers was strong and rather homogeneous, indicating similarities in terms of collagen alignment with the pure collagen film. A similar birefringence was observed for the (collagen + SiNP-PHSRN) biocomposite, with the presence of a few millimetric fibril bundles having a width of ca. 100–200 µm (yellow arrows, Figure 5b1).

The absence of clusters and the presence of fibril bundles were also observed when mixtures of SiNP-RGDS and SiNP-PHSRN or bifunctionalized SiNPs were present (yellow arrows in Figure 5c1,d1). However, for these two biocomposites, the lack of contrast between the two polarizers positioning signs for the disorganization of collagen within the films, similarly to (collagen + SiNP-SO_3_^−^) biocomposites (Figure 4c1,c2). Unexpectedly, when combining these conditions within a single bionanocomposite, i.e., (collagen + SiNP-RGDS-PHSRN + SiNP-SO_3_^−^), a rather homogeneous material could be observed, lacking nanoparticle clusters and exhibiting a strong birefringence (Figure 5e1,e2).

Overall, PLM investigations showed that a global structural alignment was reached for most of the bionanocomposites investigated: (collagen + *nf*-SiNPs), (collagen + SiNP-RGDS), (collagen + SiNP-PHSRN) and (collagen + SiNP-RGDS-PHSRN + SiNP-SO_3_^−^). Such an alignment was found to be disturbed after adding SiNP-SO_3_^−^ particles and when combining the two peptides, either on two populations of particles (SiNP-RGDS + SiNP-PHSRN) or combined on a single particle (SiNP-RGDS-PHSRN).

### 3.3. Microstructure of the Different Bionanocomposites

SEM imaging was then performed to examine the gel structure at the fiber level. Concerning the distribution of SiNPs, Figure 6 shows that nanoparticle aggregates are mainly observed in presence of *nf*-SiNPs and SiNP-SO_3_^−^ (yellow arrows), whereas, in all other nanocomposites, isolated SiNPs prevail, in agreement with PLM observations. In parallel, variations in fiber alignment between the different samples are not straightforward to identify, except for the (collagen + SiNP-RGDS-PHSRN + SiNP-SO_3_^−^) composite, where fiber alignment seems more pronounced and correlated with a larger fiber diameter compared to all composites and similar to pure collagen (Figure 6h). Finally, Figure 6i presents a representative image of a bionanocomposite cross section showing the organization of fibers perpendicular to the plan of the section.

To support these observations, fiber diameters and orientations were determined by image analysis. Concerning the diameters of fibers compared to the pure collagen film, significant decreases were measured in presence of *nf*-SiNPs, SiNP-SO_3_^−^, and when adding the two peptides, either as mixture of monofunctional SiNPs or as bifunctionalized SiNPs. (Figure 7a). Meanwhile, no significant variation in diameter compared to collagen could be observed in presence of SiNPs conjugated with a unique peptide (SiNP-RGDS, and SiNP-PHSRN) or in presence of the SiNP-RGDS-PHSRN + SiNP-SO_3_^−^ mixture. In contrast, although there was a general trend in the decrease of the average fiber orientation in the presence of particles, especially for the SiNP-RGDS and SiNP-RGDS-PHSRN + SiNP-SO_3_^−^ samples, such variations were not statistically significant, with the distribution of orientations showing a ca. 60° standard deviation in all cases (Figure 7b and see Section 2).

Overall, SEM observations confirm that playing with the surface chemistry and functionalization of SiNPs affects the structure of the resulting bionanocomposites, in terms of nanoparticle distribution, collagen fiber organization and, but to a limited extent only, fibril diameter. Further in vitro assays were then implemented to decipher the impact of collagen network structure and SiNP incorporation on the adhesion and spreading of cells.

### 3.4. Biological Behavior

The 3T3 fibroblast cells were seeded on the different bionanocomposites and their adhesion and proliferation investigated by immunostaining of actin cytoskeleton and nuclei (Figure 8). The spreading of fibroblasts on the different biomaterials, characterized by a spindle-like shape morphology with clearly visible stress fibers (Figure 8a2, white arrows), shows that those magnetically-aligned bionanocomposites are perfectly suitable for cell adhesion and spreading. Alamar blue assay revealed that only the (collagen + SiNP-RGDS-PHSRN + SiNP-SO_3_^−^) bionanocomposites induced a significantly different, and superior, metabolic activity of the cells after 24 h compared to pure collagen (Figure 8b). Cell number presented no significant difference between all biocomposites investigated but high standard deviations were noticed for several samples (Figure 8c). Finally, in terms of cell attachment, as evaluated by measurement of cell area, the (collagen + SiNP-RGDS-PHSRN + SiNP-SO_3_^−^) bionanocomposites was again the only one showing a significant and improved cell behavior (Figure 8d).

## 4. Discussion

Bionanocomposites can be advantageously designed to enhance the performances of (bio)materials by playing at the structural level and/or by providing additional functionalities. This can be typically achieved by combining biological scaffolds and biofunctionalized silica nanoparticles, as recently reported [26,29]. The processing of the materials can also be used to further tune their properties. This is the case with the use of high magnetic field that has been shown to control the alignment of fibers in collagen biomaterials [11,12,13,17,18]. Here, we have combined the magnetic alignment of collagen fibers with the incorporation of functionalized SiNPs able to modify scaffold structure and/or to confer signaling abilities to the composite materials. Figure 9 summarizes the structural features as revealed by PLM and SEM for the eight bionanocomposites investigated, together with their biological performances in terms of adhesion and spreading of fibroblasts. From these data, several points can be discussed:

(1)High magnetic field can successfully be implemented on collagen-based bionanocomposites to induce a structural anisotropy at the micron and millimeter scale after addition of silica nanoparticles. Indeed, birefringence and fiber alignment were observed for most bionanocomposites investigated in this study.(2)From a biological perspective, and very importantly, all magnetically-exposed bionanocomposites were suitable for cell adhesion and spreading as no decrease in terms of metabolic activity, cell number and attachment were observed after fibroblast seeding with neither of the composites investigated compared to the pure collagen membrane.(3)From a structural point of view, the loss of birefringence of the composites can be to some extent related to the formation of SiNP clusters, as partially observed with the formation of *nf*-SiNP clusters, and strongly evidenced in presence of SiNP-SO_3_^−^. In the latter case, the strong interaction between sulfonate groups and collagen triple helices may prevent fiber from alignment as shown by PLM.(4)Conversely, the loss of birefringence was also observed in absence of clusters (see (collagen + SiNP-RGDS + SiNP-PHSRN) and (collagen + SiNP-RGDS-PHSRN) composites). In this case, it may be attributed to the formation of large bundles of collagen fibers evidenced by PLM, which are believed to hamper alignment at the macroscale. However, local alignment of the fibers at the micrometer scale would still be possible, explaining why no difference could be evidenced for this parameter by analysis of the SEM images.

Of particular interest is the magnetically aligned (collagen + SiNP-RGDS-PHSRN + SiNP-SO_3_^−^) bionanocomposite, which appears to be the most promising candidate for biomaterials engineering. This composite combines sulfonated SiNPs—as structural cue—and SiNPs conjugated with the two integrin-binding sequences RGDS and PHSRN—as bio-chemical cue. After the application of a high magnetic field, strong birefringence and fiber alignment was observed. Moreover, while qualitatively similar to the structure of the collagen biomaterials, performances in terms of metabolic activity and cell attachment were significantly improved owing to the presence of scaffolding and signaling cues. While corresponding peptide-based collagen composites do not show improved biological properties compared to collagen alone, the combined addition of SiNP-SO_3_^−^ favor both fibroblast metabolic activity and attachment. In parallel, the scaffolding effect attributed to the presence of sulfonate alone does not secure birefringence nor does it improve cell activity either, as observed with (collagen + SiNP-SO_3_^−^) composites.

This result can be discussed in light of recent studies on the design of bionanocomposites, which incorporate SiNPs displaying scaffolding (sulfonate) and/or signaling (peptide) cues. In the first one, composite collagen threads (millimeter in diameter) were fabricated to promote neuron-like cell differentiation [26]. In this situation, the optimal biomaterial was obtained in the presence of SiNP-SO_3_^−^, while the addition of peptide-conjugated SiNP had no clear benefit. This was attributed to the structuring effect of sulfonate that induced local heterogeneities in the collagen network, creating gradients that promoted neuron cell differentiation. However, limited accessibility of the biofunctionalized particles that were mostly buried within the millimeter thread bulk prevented their interaction with cells. In the second work, nanocomposite films with a thickness ca. tens of micrometers were processed to improve peptide epitope accessibility to cells. This successfully enhanced fibroblast adhesion and spreading when incorporating biofunctionalized SiNPs similar to those used here (i.e., SiNP-RGDS, SiNP-PHSRN and SiNP-RGDS + PHSRN) [29]. In this case, the use of peptide amphiphiles as ECM building blocks instead of collagen decreased the impact of sulfonate-modified particles as no strong interaction with the matrix was expected. In this light, the present study combines (1) specific interactions between collagen and SiNP-SO_3_^−^, (2) micrometer-scale thickness of the materials for improved accessibility of conjugated peptides to cells, and (3) dual display of RGDS and PSHRN sequences. Here, these elements act in a synergetic way as the collagen network structure induced by the sulfonated nanoparticles seems to be particularly favorable to cell accessibility to the peptide-bearing ones. This has allowed the design of an optimal bionanocomposite for fibroblasts adhesion and spreading. Very importantly, this work shows that magnetic alignment can occur during collagen fibrillogenesis in presence of these multi-functional nanoparticles as revealed by SEM and PLM imaging and controlling collagen fiber alignment and the global anisotropy of the matrix respectively. From the optimization of magnetic orientation on collagen-silica bionanocomposites, future works will now focus on the impact of alignment on cells having a higher sensitivity to matrix alignment, including neurons and invading cancer cells.

## 5. Conclusions

Collagen-silica bionanocomposites can be designed as efficient ECM mimics to enhance fibroblast adhesion and spreading. This is achieved by the careful engineering of the interface of their constitutive components, and the fine tuning of their composition. With these tools in hand, scaffolding cues and multiple display of signaling peptides can be combined into a unique bionanocomposite material. Beyond all this, high magnetic field can be implemented to add an additional level of control over the global structuration of the bionanocomposites. We showed in this work that inducing collagen fibrillogenesis under magnetic field accommodate bioconjugated nanoobjects with preserved dispersion, while controlling the global anisotropic structure of the composite. This highlights that magnetic field alignment is compatible with the engineering of collagen-silica composites with improved cell performances. Managing these multiscale interplays— collagen/SiNPs; SiNPs/cells; collagen/SiNPs/magnetic field—opens news perspective for the engineering of collagen-silica composites with improved cell performance that will be investigated in the future with cells exhibiting high sensitivity to matrix alignment.

## Figures and Tables

**Figure 1 biomolecules-11-00749-f001:**
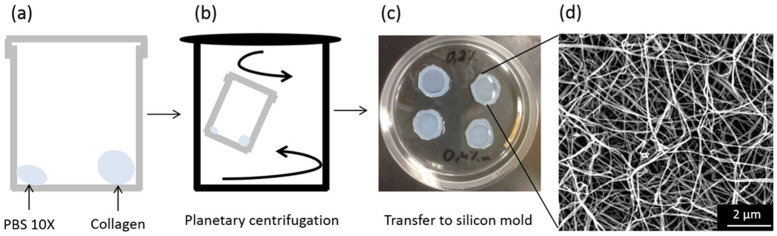
Collagen gel preparation (no magnetic alignment). (**a**) Collagen and PBS are introduced into a plastic pot (9:1, *v*:*v*). (**b**) A double rotation system allows removing air bubbles. (**c**) After transfer into a 2 mm thick silicone mold, gelling solutions are left at 28 °C for 3 h. (**d**) SEM images of collagen fibrillary networks.

**Figure 2 biomolecules-11-00749-f002:**
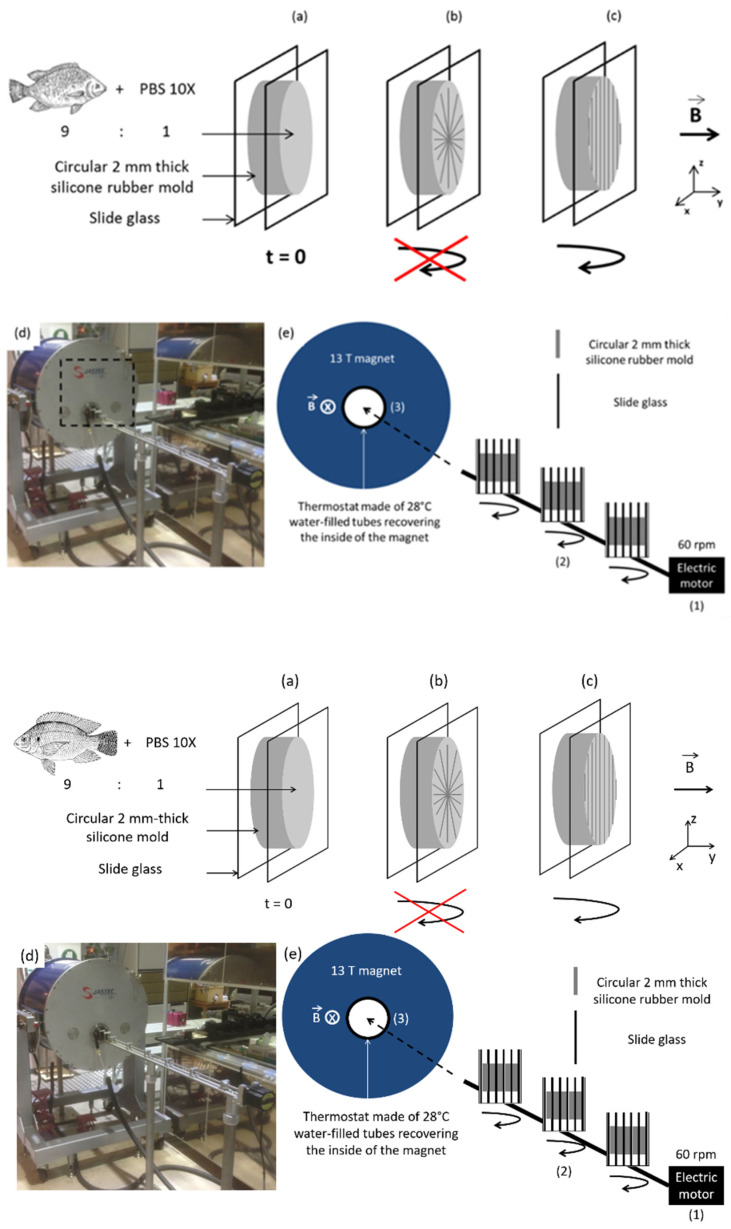
(**a**) Magnetic field application in a collagen gel. (**b**) Under a high magnetic field along y axis and without rotation, collagen fibers perpendicularly align against magnetic field (xz plan). (**c**) Upon rotation, only fibrils along the axis of rotation (z axis) remain. (**d**) Photo of the set-up, and (**e**) scheme of the essential characteristics: electric motor for stirring samples, magnetic field and thermostat.

**Figure 3 biomolecules-11-00749-f003:**
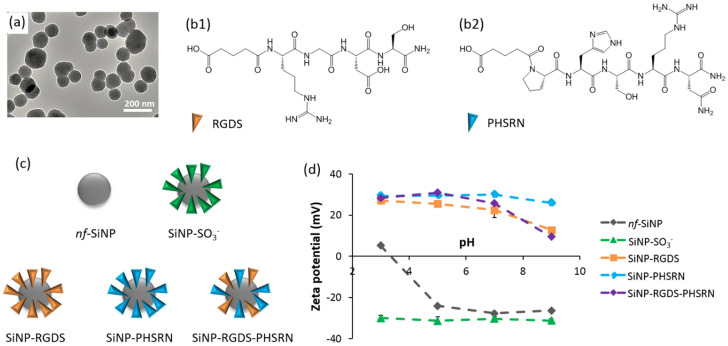
(**a**) TEM image of *nf*-SiNP. (**b1**,**b2**) Molecular structures of the RGDS and PHSRN peptides. (**c**) Scheme of the different SiNP surface chemistries. (**d**) Characterization of SiNPs by zeta potential measurements.

**Figure 4 biomolecules-11-00749-f004:**
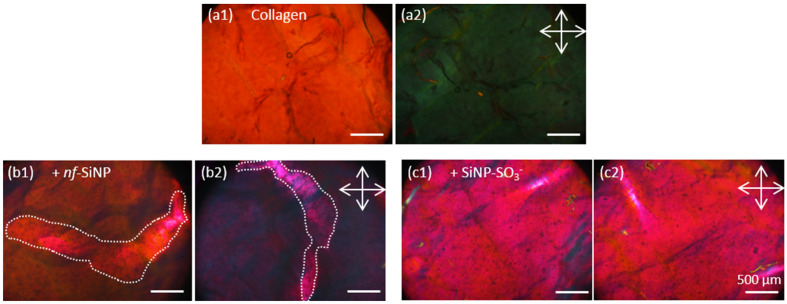
PLM images of (**a**) collagen films and composites with (**b**) *nf*-SiNP and (**c**) SiNP-SO_3_^−^ after EDC crosslinking. (**1**) and (**2**) stand for PLM imaging performed by taking 2 serial images with the sample oriented vs. the cross polarizers (white arrows) at maximal transmission (**1**) and at extinction (**2**). A waveplate was added to improve contrast.

**Figure 5 biomolecules-11-00749-f005:**
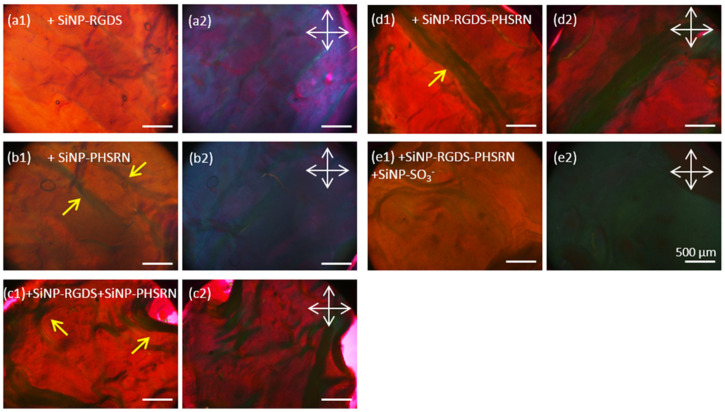
PLM photos of composites made of collagen combined with (**a**) SiNP-RGDS, (**b**) SiNP-PHSRN, (**c**) SiNP-RGDS + SiNP-PHSRN, (**d**) SiNP-RGDS-PHSRN and (**e**) SiNP-RGDS-PHSRN + SiNP-SO_3_^−^ after EDC crosslinking. (**1**) and (**2**) stand for PLM imaging was performed by taking 2 serial images with the sample oriented vs. the cross polarizers (white arrows) at maximal transmission (**1**) and at extinction (**2**). Yellow arrows show the presence of millimetric fibrils. A waveplate was added to improve contrast.

**Figure 6 biomolecules-11-00749-f006:**
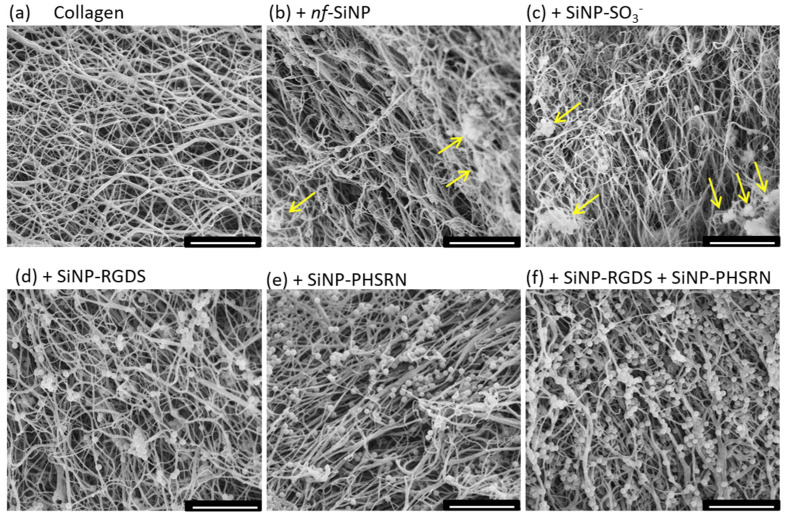
SEM images of (**a**) collagen, (**b**) + *nf*-SiNP, (**c**) + SiNP-SO_3_^−^, (**d**) + SiNP-RGDS, (**e**) + SiNP-PHSRN, (**f**) + SiNP-RGDS + SiNP-PHSRN, (**g**) + SiNP- RGDS-PHSRN, and (**h**) + SiNP-PHSRN-RGDS + SiNP-SO_3_^−^, and (**i**) cross section of the collagen + SiNP-SO_3_^−^biocomposite. Yellow arrows indicate nanoparticle aggregates. The scale bar is 5 μm for all images.

**Figure 7 biomolecules-11-00749-f007:**
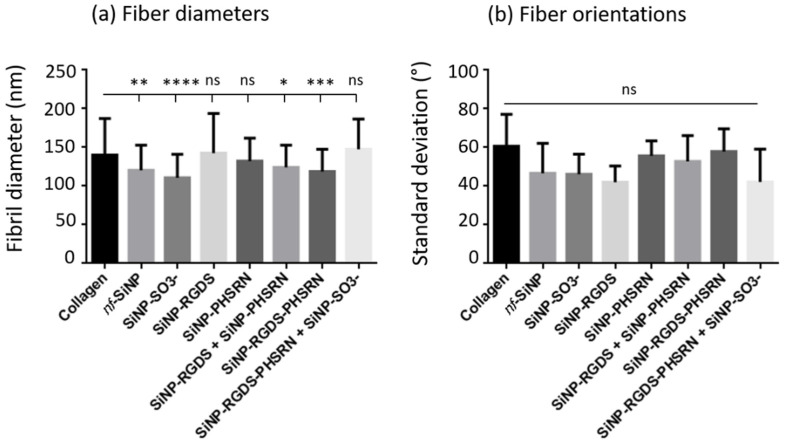
Fibril diameters and orientations measured from SEM images (performed with ImageJ software on 50 fibers for each condition). Statistical tests realized using ANOVA (Dunnett) parametric test. Each column represents mean with SEM (ns _ *non-significant*_ *p* > 0.05; * *p* ≤ 0.05; ** *p* ≤ 0.01; *** *p* ≤ 0.001, **** *p* ≤ 0.0001).

**Figure 8 biomolecules-11-00749-f008:**
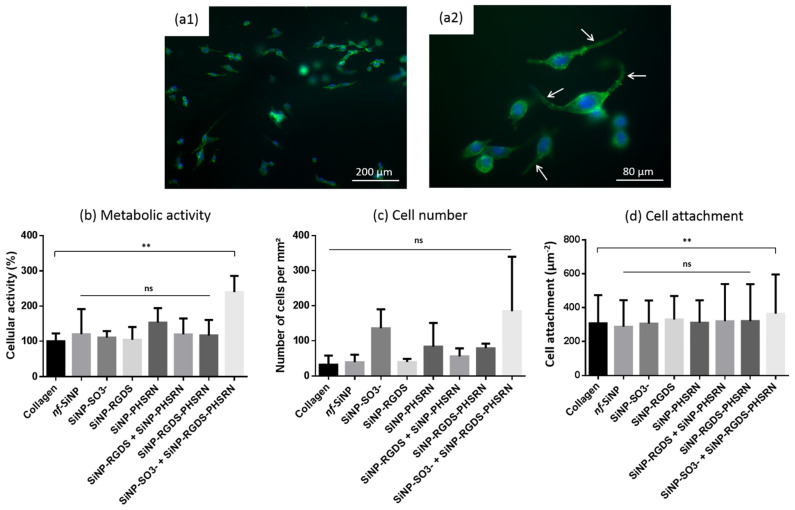
(**a1**,**a2**) Fluorescence microscopy images of 3T3 cells cultured for 24 h on magnetically-aligned collagen films. Actin (green: phalloidin) and nucleus (blue: DAPI) staining. White arrows show stress fibers. (**b**) Metabolic activity measured by Alamar blue assay. Statistical test realized using Mann-Whitney non parametric test; each condition issued from three independent experiments. (**c**) Cell number obtained by counting all cells on a 4 mm^2^ composite surface. (**d**) Cell attachment obtained by measuring the projected areas of cells. Statistical tests for (**c**,**d**) realized using ANOVA (Dunnett) parametric test. Each column represents mean with SEM (ns _ *non-significant*_ *p* > 0.05; ** *p* ≤ 0.01).

**Figure 9 biomolecules-11-00749-f009:**
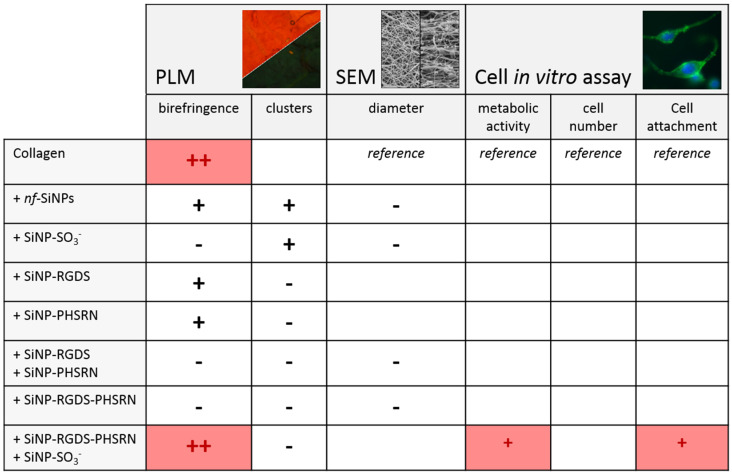
Structural features revealed by PLM and SEM and biological performances from cell in vitro assays for the eight bionanocomposites investigated in this study. The boxes left empty correspond to non-significant results compared to pure collagen films.

## Data Availability

The data presented in this study are available on request from the corresponding author.

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
