# Peer review of "Magnetic Field Alignment, a Perspective in the Engineering of Collagen-Silica Composite Biomaterials"

_biomolecules, 2021, doi:10.3390/biom11050749_

Round 1

Reviewer 1 Report

The authors report an interesting study in which they used a magnetic field to fabricate a bionanocomposite composed of aligned collagen nanofibrils containing multifunctional silica nanoparticles.  The surface chemistry of the silica nanoparticles was varied and the cell response was studied.

The authors provide an interesting approach to designing tissue scaffolds with morphology that mimics the ECM and multifunctionality by incorporating the silica nanoparticles. The following points should be considered:

  • On lines 29a and 295, it is stated that the presence of black spots can be attributed to the presence of silica particle aggregates, however the wording of line 294 is a bit confusing. If the particles are aggregated, then they are not homogeneously dispersed.  It would be better to delete the words homogenously dispersed on line 294. 
  • On line 327, alignment should be used instead of anisotropy.
  • On line 330, alignment should be used instead of anisotropy.

Reviewer 2 Report

Manuscript ID: biomolecules-1203544

Title: Magnetic file-assisted electrospinning of aligned straight and wavy polymeric nanofibers

Authors: Liu Y, Zhang X, Xia Y, Yang H

_____________________________________________________________________________________

The authors have made an interesting work. They have well designed and executed the experiments, I recommend this article for publication after revising the manuscript based on the following queries.

On Page 3, section 2.1.4.: explain the ultrasonication conditions and the instrument details.

On Page 4: 2.3. Preparation of the bionanocomposite: the details of incorporation of functionalized SiNPs with collagen with stoichiometric ratios need to be elaborated.

The alignment and arrangement of collagen fibers under different magnetic fields can be studied. The authors can justify the use of 13T magnetic fields for the preparation of bionanocomposites.

SEM cross-section of composite materials can be provided along with Figure 6 to check the alignment. Mention the yellow arrows in the figure legend.

Make a separate section of materials used in the experiment under section 2. Materials and methods.

_____________________________________________________________________________________
